# Successful Endovascular Surgery for Iatrogenic Common Iliac Artery Injury during Lumbar Spine Surgery: A Case Report

**DOI:** 10.3390/medicina58070927

**Published:** 2022-07-13

**Authors:** Chien-Ming Chin, Kuan-Lin Liu, Ing-Ho Chen

**Affiliations:** 1Department of Orthopedics, Hualien Tzu Chi Hospital, Tzu Chi University, Buddhist Tzu Chi Medical Foundation, Hualien 970, Taiwan; addison30912@gmail.com (C.-M.C.); liulob@gmail.com (K.-L.L.); 2Sports Medical Center, Hualien Tzu Chi Hospital, Hualien 970, Taiwan; 3Department of Orthopedics, School of Medicine, Tzu Chi University, Hualien 970, Taiwan

**Keywords:** endovascular surgery, iatrogenic common iliac artery injury, lumbar spine surgery, case report

## Abstract

An 80-year-old man was admitted with an L5 compression fracture, L4/5 spondylolisthesis, and L5 radiculopathy and underwent a TLIF procedure. Refractory hypotension occurred, though it indicated a possible great vessel injury with vasopressor and fluid infusion. Emergent intraoperative angiography was performed, which showed extravasation at the right common iliac artery. Resuscitative endovascular balloon occlusion of the aorta followed by right common iliac artery stenting was successfully performed to arrest the bleeding. The iatrogenic right common iliac artery laceration was complicated with abdomen compartment syndrome and acute kidney injury. The patient received supportive care, including continuous venovenous hemofiltration (CVVH) for a week, after which the patient’s condition improved. The patient did not have any residual complications at the one-month follow-up. Great vessel injury during the TLIF procedure is rare but fatal. Refractory hypotension is indicative of a great vessel injury. Endovascular intervention is a fast and promising method to diagnose and treat arterial injury.

## 1. Introduction

Transforaminal lumbar interbody fusion (TLIF) is a common, effective and safe treatment to manage spondylolisthesis with neurological deficit. Iatrogenic vessel injury during a fusion bed preparation is rare but a potentially fatal complication. Here, we introduce a case where an iatrogenic right common iliac artery injury occurred during TLIF. The vessel was resuscitated by endovascular balloon occlusion in the aorta to repair the vascular lesion.

## 2. Illustrative Case

An 80-year-old man was admitted to our hospital due to low back pain with radiating numbness and soreness in the right buttock and lateral leg. He had been complaining about these symptoms for two years, and they worsened after a falling episode. The patient was brought to the orthopedic department, where a physical examination showed positive lumbar (L) 4/5 knocking tenderness with right L5 radiculopathy. Lumbar spine X-ray and magnetic resonance imaging (Figure 1A–C) showed an L5 body compression fracture, L4/5 spondylolisthesis, and right L4 lateral recess stenosis and severe central stenosis. Based on the diagnosis of an L5 compression fracture with L4/5 instability and neural compression, L4/5 posterior instrumentation fusion and TLIF were performed. The surgery was performed by the open surgical approach. During the preparation of the L4/5 fusion bed, accidental penetration with a rasp occurred while removing a soft callus from the L5 upper endplate. The surgical site was clean, and no active bleeding was observed. However, progressive tachycardia and hypotension were noted five minutes later. The gauze was packed into the posterior wound, and the patient was rapidly moved to the supine position. A distended abdomen was noted. Thereafter, pulseless electric activity was noted, and cardiopulmonary resuscitation was performed. Additionally, retroperitoneum fluid accumulation was observed on sonography. An emergency angiogram showed extravasation of contrast from the right common iliac artery (Figure 2A). An endovascular procedure was performed instead of an exploration laparotomy due to the lesion site’s proximity to the vertebral body and difficulty in achieving a direct approach for repair.

The endovascular procedure began with resuscitative endovascular balloon occlusion of the aorta (REBOA) at the abdomen aorta level to stop the bleeding. A stent graft was placed in the injured right common iliac artery (Figure 2B,C). After the endovascular procedure, the hemodynamic status became stable. The patient was then shifted to a right decubitus position, and the rods and bone substitute were inserted to complete the spinal surgery.

In the intensive care unit, the patient developed acute abdomen compartment syndrome with intra-abdominal pressure (IAP) (20 mmHg), measured indirectly using the bladder technique. Nasogastric tubes were inserted for intestinal pressure decompression. IAP, lactate, liver enzyme, and creatinine levels were closely monitored as a reference to decide when it was appropriate to perform an exploratory laparotomy.

Post-operative oliguria and an increased creatinine level confirmed acute kidney injury (AKI), which may have been caused by hemorrhagic shock or abdominal compartment syndrome. The AKI did not resolve as expected; however, the shock status of the patient improved without inotropic agent usage—the IAP level did not increase, and lactate and liver enzyme levels decreased (Table 1). Therefore, we performed bedside abdomen echocardiography which revealed that a hematoma compressed the bilateral renal arteries. Based on this finding, continuous venovenous hemofiltration (CVVH) was initiated on postoperative day (POD) 8 to provide temporary renal support for the AKI. After a week of CVVH therapy, renal function gradually recovered even though the retroperitoneal hematoma surrounding the renal arteries and kidney was detected on a CT scan (Figure 3A,B). The patient was discharged 31 days postoperatively without renal, bowel, or neurologic sequelae. A month after discharge, the patient followed up at an outpatient clinic. Back pain and neurologic symptoms had improved. His renal function returned to its preoperative condition. His abdomen was soft and free of tenderness. A follow-up abdominal CT showed total resorption of the retroperitoneal hematoma and no evidence of pseudoaneurysm and arteriovenous (AV) fistula. (Figure 3C).

## 3. Discussion

Iatrogenic vascular injury during lumbar spine surgery is a significant complication because it is difficult to manage and potentially fatal [1,2]. Although the incidence of major vascular injury is rare (0.01–0.05%) [2,3], the mortality rate could reach high at 10–65% [2,4]. According to Papadoulas et al. [2], the incidence of AV fistula, laceration, and pseudoaneurysm due to spine surgery-related vascular injury is 66%, 33%, and 3%, respectively. The three most common clinical presentations of vascular laceration are hypotension (77%), active bleeding (53%), and distended abdomen 20% [2,5]. The symptoms of these complications may present intraoperatively or during the perioperative period. The clinical presentations of AV fistula and pseudoaneurysm can be chronic. AV fistulas usually mimic congestive heart failure with accompanying leg edema. Pseudoaneurysms usually present as new-onset lumbar back pain or neurological symptoms [2,4,6]. They could be diagnosed up to months or years after operation and treated in an elective setting [7,8].

In contrast, most symptoms of vascular laceration may present intraoperatively and should be tended to immediately; on some occasions, the prone position in which the patients are operated on can cause a degree of vascular compression that can temporally tamponade the vascular tears, causing such tears to go unnoticed, thus highly clinical suspicion is required during the perioperative period [9]. Our patient’s tachycardia and hypotension progressed rapidly, and pulseless electrical activity was detected. Later, the right common iliac artery laceration was detected by angiography, the diagnostic tool for detecting vascular injuries. This process is compatible with that described in the previous literature [2,4,10].

Treatment for such vascular injuries can be performed through open or endovascular surgery. Table 2 lists some advantages and disadvantages of open versus endovascular treatment for vascular injuries related to spine surgery and also gives a literature review of the reported cases regarding the injured vessel, injured type, diagnosis timing, and mortality. Open surgery enables the control of acute bleeding and has been traditionally described and recommended to repair these lacerations [10,11].

Shih et al. described a case that used laparotomy with the abdomen aorta clamped at the level of the celiac axis. However, massive retroperitoneal hematoma complicated the vascular repair. Although the injury to the vessel was repaired, the patient died in the operating room [12]. For our patient, we did not choose open surgery for the following reasons. First, when performing an open laparotomy, compartment pressure is released. This can cause the wound to active bleed, further complicating the vessel repair. Second, the lesion site of the great vessel that adhered anteriorly to the vertebra might have made the repair difficult. Third, since a vein was close to the aorta, the concomitant venous injury was possible. Lastly, an open laparotomy may have caused intestinal exposure and organ swelling, making the abdominal wall difficult to close, contributing to perioperative morbidity and mortality rates [13].

Zajko et al. first reported endovascular repair in 1995. Using a stent graft, they successfully treated a common iliac artery-to-inferior vena cava fistula [14]. The endovascular approach would avoid the above-mentioned disadvantages of the open method. Since then, endovascular stenting repair techniques have been increasingly applied to vascular injuries following lumbar spine surgery. However, almost all of them were applied in postoperative cases (e.g., AV fistula and pseudoaneurysm) [8,15,16,17,18] (Table 2). This technique is rarely applied in vascular laceration due to a lack of availability of the related personnel and facilities in an emergent setting [10]. For this reason, the current literature suggests that primary suture should be considered a first-line treatment [6,14,15,19].

Before the placement of the stent, REBOA is first performed during the same endovascular procedure. In REBOA, a balloon occludes the artery over or above the injury level to help achieve hemostasis, control initial bleeding, and stabilize blood pressure. This technique has made endovascular repair easier, safer, and faster [20]. Although the endovascular approach might salvage the devastating complication of large vessel laceration, there are limitations to this technique. First, the personnel and facilities should be available in a short time. Second, the risk of stenosis, thrombogenesis, and long-term use of anti-thrombotic medication is inevitable.

The abdominal compartment pressure due to the massive hematoma was not relieved after endovascular repair. However, the necessity of surgical decompression was douted [21,22]. In this case, close monitoring of abdominal pressure, optimization of systemic perfusion, indwelling nasogastric tube for gastrointestinal and colonic decompression, and temporary continuous renal replacement are successful strategies we strongly recommend [23].

## 4. Conclusions

Iatrogenic vessel injury due to TLIF is a major complication that increases perioperative morbidity and mortality. A sudden occurrence of refractory hypotension indicated a great vessel injury. Endovascular surgery is a promising approach to control bleeding and treating vascular injury. 

## Figures and Tables

**Figure 1 medicina-58-00927-f001:**
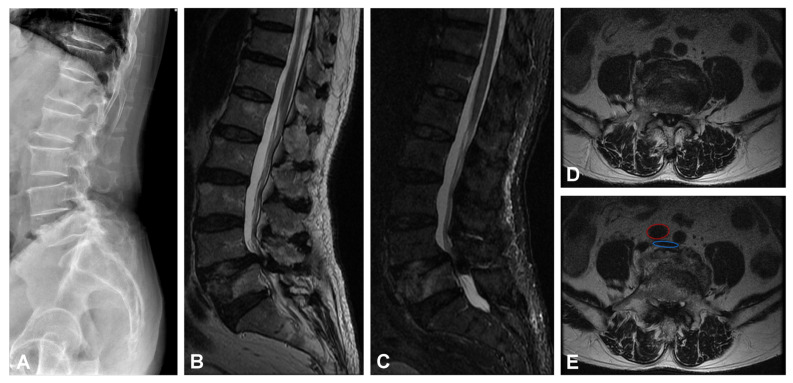
Images of preoperative. (**A**) Lateral X-ray view of the lumbar spine. (**B**) Sagittal T2EI FSE MRI and (**C**) sagittal STIR MRI L5 old compression fracture, L4/5 spondylolisthesis. (**D**,**E**) Axial T2WI FSE MRI showed L4/5 lateral recess stenosis and severe central stenosis. Images (**E**) showed the pre-operative vascular position of the L5 vertebral body, which is indicated by a red circle symbol (right common iliac artery) and a blue circle symbol (inferior vena cava around the bifurcation).

**Figure 2 medicina-58-00927-f002:**
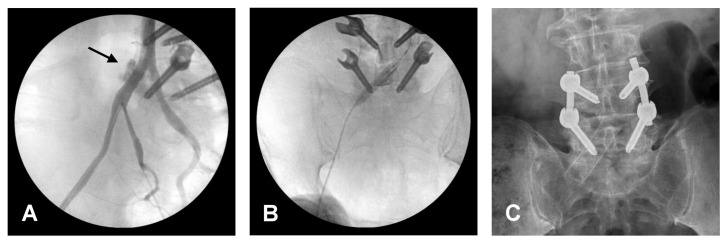
Intra-operative angiography and post-operative X-ray. (**A**) Contrast extravasation vias the right common iliac artery, which is indicated by a black arrow. (**B**) A stent was placed to repair the laceration. (**C**) Demonstrate stent insertion.

**Figure 3 medicina-58-00927-f003:**
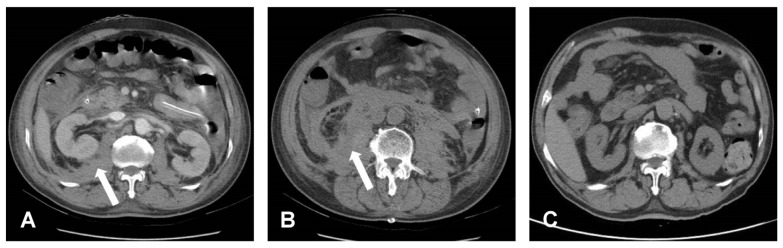
Post-operative CT. (**A**,**B**) Bilateral retroperitoneal hematoma, especially at the right side, which is indicated by a white arrow symbol. (**C**) Post-operative three months showed totally resolution of retroperitoneal hematoma.

**Table 1 medicina-58-00927-t001:** Pateint’s Laboratory Data during Hospitalization.

	Pre-OP	POD 1	POD 2	POD 3	POD 4	POD 5	POD8	POD15	POD31
BUN (mg/dL)	13	18	31	40	50	54	93	85	24
CRE (mg/dL)	1	2	3.5	5.2	5.6	6	7	4.8	1.2
eGFR (mL/min)	76.22	34.25	17.96	11.37	10.44	9.64	8.07	12.47	61.76
Lactate (mmol/L)	0	11.8	1.4	1.7	1.1	0.6			
AST (U/L)	19	155	177	316	160	66			
ALT (U/L)	22	109	109	207	134	73			

**Table 2 medicina-58-00927-t002:** Advantages and disadvantages of open versus endovascular treatment for vascular injuries related to spine surgery and literature review of previous cases.

	First Author	Year	Case No.	ACS	Injured Vessel	Injured Type/Diagnosis Timing	Clinical Outcome
**Open repair** Advantage Direct repair. No need for long-term anti-thrombotic agent use. Disadvantage Bleeding might become active, complicating the vessel repair. The lesion site of the great vessel that adhered anteriorly to the vertebra made the repair complex.	Papadoulas S et al. [2]	2002	29	N/A	Aorta 7 IVC 2 CIA 15 CIV 1	Laceration/ 28 cases were diagnosed intraoperatively or within 24 h	Total mortality rate 20%; 38% in aortic laceration
			55	N/A	CIV	AV fistula/ Most patients were diagnosed postoperatively one month to one year	Mortality rate 5%
			2	N/A	CIV	Pseudoaneurysm	Mortality rate 0%
	Jung HS et al. [6]	2017	4	N/A	CIA	Laceration/3 cases were diagnosed intraoperatively; the other was diagnosed on the day of the operation	Survival
	Szolar DH et al. [10]	1996	1	N/A	CIA	Laceration/ postoperatively	Hemorrhagic infarction, died with stroke-related complication
			1	N/A	CIA	AV fistula/ 19 days after spine surgery	N/A
			2	N/A	Aorta CIA	Pseudoaneurysm/1 case was diagnosed intraoperatively, the other case was diagnosed 11 months after the operation	survival
	Shih et al. [12]	2020	1	yes	Abdominal aorta	Laceration/ Patient was diagnosed in postoperative room	Intraoperative cardiac arrest
	Boyd et al. [7]	1965	2	N/A	Aorta CIA	Laceration/ intraoperatively	One patient encountered intraoperative cardiac arrest One patient died after post-op 3 months
			2	NA	CIA	AV fistula/ One case was diagnosed postoperative 4 months; one case was diagnosed post-operative 4 years	Survival
**Endovascular** Advantage Minimal invasive With REBOA technique, quicker, easier and safer Disadvantage Need well-experienced endovascular surgeon Facility available Long-term anti-thrombotic agent use Risk of stenosis Thrombogenesis	Papadoulas S et al. [2]	2002	1	N/A	CIV	Pseudoaneurysm	Mortality rate 0%
	Jung HS et al. [6]	2017	1	N/A	CIA	AV fistula/ 60 months after spine surgery	Survival
			2	N/A	CIA	Pseudoaneurysm/ 1 case was diagnosed 2 weeks after spine surgery, while the other was 23 months after spine surgery	Survival
	Canaud L et al. [9]	2011	3	N/A	Aorta CIA	Laceration/ intraoperatively	Survival
			2	N/A	CIA	AV fistula/ Postoperatively, not mentioned clear time	Survival
			2	N/A	CIA	Pseudoaneurysm/ Postoperatively, not mentioned precise time	Survival
	Shih et al. [12]	2020	1	yes	CIA	Laceration/ intraoperatively	survival
	Momoh et al. [8]	2008	1	N/A	Aortic	Pseudoaneurysm/ 2 months postoperatively	Survival
	Hong, Seong J [15]	2000	1	N/A	CIA	Pseudoaneurysm/ Post-operative day 5	Survival
	Park et al. [17]	2013	1	N/A	CIA	Pseudoaneurysm/ Post-operative day 1	Survival

ACS—abdomen compartment syndrome; IVC—inferior vena cava; CIA—common iliac artery; CIV—common iliac vein.

## Data Availability

Data sharing not applicable.

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
