# Peer review of "Successful Endovascular Surgery for Iatrogenic Common Iliac Artery Injury during Lumbar Spine Surgery: A Case Report"

_medicina, 2022, doi:10.3390/medicina58070927_

Round 1

Reviewer 1 Report

 The authors reported a valuable case of right common iliac artery injury during TLIF that was successfully repaired using endovascular balloon occlusion of the aorta and saved the patient's life.

However, I think that there are several concerns that should be made before publication.

 As the authors point out, endovascular treatment is common in cases of large vessel injury during spine surgery, making this paper less novel

Please review more literature.

A little more information on the timing (intraoperative and postoperative cases) and occasion of diagnosis of vascular injuries related to spine surgery would make the paper more informative!

 I would like a summary table of open versus endovascular treatment for vascular injuries during spine surgery (year, advantages, disadvantages/complications, Survival rate etc.)

Author Response

Responses to Comments from Reviewers

We would like to thank the reviewers for their extensive assessment of our manuscript, and for important and helpful comments and suggestions. We have taken all the remarks into account, in a manner that is described in detail below together with our responses to comments. We have responded to all the reviewer’s comments in a point-by-point fashion and have revised the manuscript accordingly. The revised portions are indicated by track changes. We think that, following these suggestions, our manuscript has gained in clarity and hope that the changes made will be considered satisfactory.

Reviewer #1

Comment 1: As the authors point out, endovascular treatment is common in cases of large vessel injury during spine surgery, making this paper less novel

Please review more literature

Response:

We thank the reviewer for allowing us to explain further regarding the issue of endovascular treatment. Actually, the endovascular treatment for large vessel injury during spine surgery is rare, not to mention with intraoperative repair. In response to the reviewer’s comment, we have added a new table (table 2) in the revised manuscript to give a literature review of the reported cases. We have also modified the paragraph in the discussion section (Page 6) to explicitly point out this fact and to address this issue. 

These statements (Page 6, lines 10-14) read as: “Treatment for such vascular injuries can be performed through open or endovascular surgery. Table 2 lists some advantages and disadvantages of open versus endovascular treatment for vascular injuries related with spine surgery and also gives a literature review of the reported cases regarding the injured vessel, injured type, diagnosis timing, and mortality. As shown, the endovascular treatment for large vessel injury during spine surgery is rare, not to mention with intraoperative repair.”

Also, other statements read as: “However, almost all of them were applied in postoperative cases (e.g., AV fistula and pseudoaneurysm) [18-22] (Table 2). Only rarely was this technique applied in vascular laceration due to a lack of availability of the related personnel and facilities in emergent setting [10]. For this reason, current literature still suggests that primary suture should be considered as first line treatment [6,11,19,23].”  

Comment 2: A little more information on the timing (intraoperative and postoperative cases) and occasion of diagnosis of vascular injuries related to spine surgery would make the paper more informative!

Response: We thank the reviewer for the excellent suggestion. In response to this suggestion, we have added a new table (table 2) in the revised manuscript to give a literature review of the reported cases regarding the injured vessel, injured type, diagnosis timing, and mortality.

The part of discussion relevant to this table (Page 5, lines 14-23) read as: “According to Papadoulas et al. [2], the incidence of AV fistula, laceration, and pseudoaneurysm due to spine surgery-related vascular injury is 66%, 33%, and 3%, respectively. The three most common clinical presentations of vascular laceration are hypotension (77%), active bleeding (53%), and distended abdomen 20% [2,5]. The symptoms of these complications may present intraoperatively or perioperative period. The clinical presentations of AV fistula and pseudoaneurysm can be chronic. AV fistulas usually mimic congestive heart failure with accompanying leg edema. Pseudoaneurysms usually present as new onset lumbar back pain or neurological symptoms [2,4,6]. They could be diagnosed up to months or years after operation and treated in elective setting [17,18].”

Comment 3: I would like a summary table of open versus endovascular treatment for vascular injuries during spine surgery (year, advantages, disadvantages/complications, Survival rate etc.)

Response: We thank the reviewer for the excellent suggestion. In response to this suggestion, we have added a new table (table 2) in the revised manuscript to give a literature review of the reported cases regarding the year, advantages, disadvantages, injured vessel, injured type, diagnosis timing, and mortality. We have added statements in the discussion section to discuss the contents in the table 2 (page 6, lines 11-14).

Reviewer 2 Report

Successful Endovascular Surgery for Iatrogenic Common Iliac Artery Injury during Lumbar Spine Surgery: A Case Report

The presented case report demonstrates a fatal complication during TLIF instrumentation. Furthermore, the authors descript the laceration of the right iliac artery, which were detected by hypotension and tachycardia of the patients, in detail. 

The picture with the angiography with the blood extravasation from the right iliac artery is impressive. 

This fatal complication during the operative procedure of TLIF surgery is well known, but this report of the author’s is awesome detailed.

I will complement the involved surgeons for the good management of this fatal complication. The presentation was detailed and honestly. The Discussion part was substantial and critical. I have no points for revision.  

Therefore, I recommend the publication of this case report in Medicina. 

Author Response

Responses to Comments from Reviewers

We would like to thank the reviewers for their extensive assessment of our manuscript, and for important and helpful comments and suggestions. We have taken all the remarks into account, in a manner that is described in detail below together with our responses to comments. We have responded to all the reviewer’s comments in a point-by-point fashion and have revised the manuscript accordingly. The revised portions are indicated by track changes. We think that, following these suggestions, our manuscript has gained in clarity and hope that the changes made will be considered satisfactory.

Reviewer #2

Comment 1: The presented case report demonstrates a fatal complication during TLIF instrumentation. Furthermore, the authors descript the laceration of the right iliac artery, which were detected by hypotension and tachycardia of the patients, in detail. The picture with the angiography with the blood extravasation from the right iliac artery is impressive.

This fatal complication during the operative procedure of TLIF surgery is well known, but this report of the author’s is awesome detailed.

I will complement the involved surgeons for the good management of this fatal complication. The presentation was detailed and honestly. The Discussion part was substantial and critical. I have no points for revision. Therefore, I recommend the publication of this case report in Medicina.

Response: We thank the reviewer for the positive feedback .

Round 2

Reviewer 1 Report

The author has adequately addressed the points raised. However, table1 is missing.

Author Response

Comment 1:

The author has adequately addressed the points raised. However, table1 is missing.

Response:

We thank the reviewer for the suggestion. In response to this suggestion, we have added the missing table (table 1) about patient's basic data in the revised manuscript
